# PI-FL: Personalized and Incentivized Federated Learning

## Abstract

Existing incentive solutions for traditional Federated Learning (FL) only consider individual clients' contributions to a single global model. They are unsuitable for clustered personalization, where multiple cluster-level models can exist. Moreover, they focus solely on providing monetary incentives and fail to address the need for personalized FL, overlooking the importance of enhancing the personalized model's appeal to individual clients as a motivating factor for consistent participation. In this paper, we first propose to treat incentivization and personalization as interrelated challenges and solve them with an incentive mechanism that fosters personalized learning. Second, unlike existing approaches that rely on the aggregator to perform client clustering, we propose to involve clients by allowing them to provide incentive-driven preferences for joining clusters based on their data distributions. Our approach enhances the personalized and cluster-level model appeal for self-aware clients with high-quality data leading to their active and consistent participation. Through evaluation, we show that we achieve an 8–45% test accuracy improvement of the cluster models, 3–38% improvement in personalized model appeal, and 31–100% increase in the participation rate, compared to a wide range of FL modeling approaches, including those that tackle data heterogeneity and learn personalized models.

## 1 Introduction

Training high-quality models using traditional distributed machine learning requires massive data transfer from the data sources to a central location, which raises various communication, computation, and privacy challenges. In response, Federated Learning (FL) [1–4] has emerged as a solution to train models at the source, reducing privacy issues and addressing the need for high-quality models. However, the success of FL relies on resolving various new challenges related to statistical heterogeneity [5–10], scheduling [11–13], and incentive distribution [14–18]. Recent works have focused on training personalized models [9, 19–22] to overcome data heterogeneity challenges.

Among personalized Federated Learning (pFL) techniques, similarity-based approaches that use clustering of clients at the aggregator have gained popularity [23–27]. These personalization solutions fulfill the primary goal of overcoming data heterogeneity for specific cases. However, existing pFL solutions do not include any incentive mechanism, which is crucial in FL to motivate participants to contribute their data and computation resources. Existing incentive mechanisms [14, 16, 28] for traditional FL cannot be applied to pFL techniques because they only consider the performance contribution of clients towards training a single objective. In contrast, clients in pFL can be contributing towards multiple objectives simultaneously [7, 8, 24, 25, 29, 30]. Furthermore, traditional incentive solutions only provide monetary benefits and do not consider increasing personalized models' appeal as an incentive for encouraging active and reliable participation of clients. Without

incentives, participants may provide low-quality data [14, 16, 18] or opt-out from participation[1] [31], leading to poorly performing pFL models [10, 32, 33], as shown with empirical evaluations in the later section. Collaboration fairness [34, 35] can also be ensured by appropriately rewarding contributions and accounting for data heterogeneity [18, 36].

In addition, since existing pFL techniques assume voluntary and consistent participation from clients, the aggregator controls the client selection and training with limited knowledge of clients' training capacity, availability, frequency of new incoming data, clustering preferences, and performance requirements from the trained personalized models. These factors can directly influence the motivation of self-conscious clients to participate consistently. Our evaluation shows that this causes frequent opt-outs from uninterested clients due to uninformed clustering decisions by the server and low personalized model appeal (PMA)[2], which leads to reduced pFL performance. We also show that solving personalization and incentivization as interrelated challenges yield better outcomes for pFL than solving them as separate problems. However, this requires new paradigms for clustered pFL using data distribution information available to clients via their preferences and designing incentive mechanisms for increasing pFL appeal to reduce client opt-outs.

In this paper, we propose PI-FL that combines clustering-based pFL with token-based incentivization. Unlike previous works that control clustering from the server side, PI-FL allows clients to estimate the importance of each cluster and send their preferences for joining them to the aggregator as bids. To identify a cluster's importance to a client we use the importance weight of the cluster model as defined by FedSoft [25]. Clients also use the importance weights to perform weighted local aggregation for single-shot personalization. This client-driven clustering approach results in accurate clustering because clients can attain a global perspective from their own local dataset which is only accessible to them and the importance weights information of each cluster. This allows them to make informed decisions that the server cannot make, resulting in improved PMA and reduced opt-outs. To incentivize clients for consistent participation, PI-FL motivates clients to join clusters with the clients that are most similar to them, maximizing their contribution to the cluster and, in turn, their rewards. Good quality cluster-level models then produce more appealing personalized models for each client. The incentive mechanism treats clients as both providers and consumers. As a consumer, the client tries to attain a certain level of personalized model appeal, so it pays the provider to spend resources to participate in training for the said model in each round. Whereas as a provider, the client earns a profit based on its marginal contribution to training the cluster models. The marginal contribution is calculated with a Shapley Value approximation due to the large computational overhead of the original algorithm [38–41].

**Contributions.** Existing pFL solutions fail to include PMA as an incentive to maintain consistent participation, resulting in increased opt-outs. To address this issue, we propose PI-FL as the first contribution, which provides contribution-based incentives to achieve collaborative fairness and maintain the cluster-level and personalized models' appeal for clients to prevent opt-outs. Additionally, PI-FL has the added advantage of creating personalized models for unseen clients with unknown data distributions that perform similarly to seen clients without the need for training. Secondly, we provide theoretical analysis and empirical verification of the benefits of including incentives with personalization. Lastly, we empirically evaluate the performance of PI-FL and other pFL models.

## 2   Related work

**Cluster-based pFL:** Among the cluster-based pFL works most related to PI-FL are FedSoft [25], FedGroup [24], and [29]. FedSoft utilizes soft clustering on the basis of matching data distributions in clients with cluster models while FedGroup quantifies the similarities between clients' gradients by calculating the Euclidean distance of decomposed cosine similarity metric and [29] finds the optimal personalization-generalization trade-off from the cluster model by solving a bi-level optimization problem. This work incurs clustering overhead at each iteration and does not consider the overlap of distribution between clients wherein each client is restricted to one cluster for each training round. Other cluster-based pFL models include IFCA [42] which proposes a framework for loss-based clustering of clients and  [23] which proposes three approaches for personalization using clustering, data interpolation, and model interpolation.

**Other pFL models:** Some pFL models propose meta-learning techniques that provide methods for rapid training of a personalized model. These include fine-tuning methods such as Per-FedAvg [43]

---

[1]By "opt-out" we mean the clients voluntarily leave FL due to the lack of incentivization.

[2]Akin to global model appeal [37], we propose a new metric to measure the personalized model appeal.

and regularization of local models [44, 45]. Others works [8, 46] including FedALA [6], Ditto [30] and pFedMe [47] propose multi-task learning and model-interpolation [48] pFL models. FedFomo [7] suggests an adaptive local aggregation approach for personalization. FedProx [5] proposes a proximal term to improve the stability of FL. As per our knowledge, all of these pFL works lack qualities for attracting or sustaining long-term participation from self-conscious clients leading to an increase in opt-outs and low PMA. Moreover, most of these works require either require further training or re-clustering to adapt the personalized models for new incoming clients.

**Incentivized FL:** FAIR [14] integrates a quality-aware incentive mechanism with model aggregation to improve global model quality and encourage the participation of high-quality learning clients. FedFAIM [18] proposes a fairness-based incentive mechanism to prevent free-riding and reward fairness with Shapley value-based client contribution calculation. [31] proposes an approach based on reputation and reverse auction theory which selects and rewards participants by combining the reputation and bids of the participants under a limited budget. [16] proposes an approach where clients decide whether to participate based on their own utilities (reward minus cost) modeled as a minority game with incomplete information. Other incentivized FL works include [15, 17, 34, 49, 50]. All of these works propose standalone solutions to attract clients, however, none of them fulfill the design requirements to be used with any pFL models.

**Why existing incentive mechanisms cannot be applied directly to pFL frameworks?**

Existing FL incentivization schemes designed for motivating clients to contribute to a single global goal [14, 16, 18, 31] may not be applicable to pFL frameworks due to the multi-dimensional goals and objectives involved. In pFL frameworks, multiple objectives must be optimized simultaneously, such as cluster and personalized models per client in cluster-based pFL [24, 25, 29] or global and local models per client in multi-task learning [20, 30, 45, 47]. To encourage clients to contribute towards the multiple objectives in pFL frameworks, new incentive mechanisms need to be developed that are specifically tailored to their multi-objective nature. PI-FL uses clustering for pFL wherein the clusters memberships are changed after every $R$ training rounds. PI-FL is different from these as it forms clear boundaries between multiple cluster models and improves shared learning between cluster similarities through multiple participation at the client level. PI-FL incorporates maintaining PMA for consistent client participation with an incentive mechanism that directly motivates personalized training on the basis of Individual Rationality (IR) constraint of game theory [14, 51].

## 3 Proposed Methodology

In this section, we introduce PI-FL, which has three main modules: the profiler, the token manager, and the scheduler as shown by the architecture diagram in Figure 1. The profiler calculates and maintains the history of client contributions using Shapley Values approximation (lines 24-27) of Algorithm 1. The profiler also aids the scheduler in forming clusters using two different modes further explained in section 3.1. The token manager orchestrates transactions, holds auctions, deducts payments, and distributes rewards as given in lines (13 and 14). The scheduler selects clients based on bids and contributions, grouping them for improved homogeneity shown in lines (20 and 27-29). Individual clients calculate the importance weights of each aggregated cluster model and send their preference bids to the Token Manager for joining clusters as shown in lines (23-28) in Algorithm 2. Clients also generate a single-shot personalized model, shown in line 29. We assume that each client will look to maximize their profits according to the principle of Individual Rationality (IR) [10, 52] and this will lead them to choose clusters in which they can contribute the most for maximum reward.

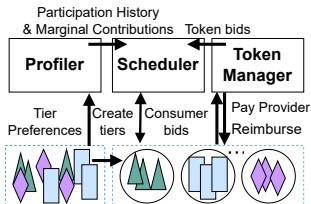

Figure 1: PI-FL design

### 3.1 Profiler

At the start of pFL training, the scheduler module forms the initial clusters by randomly assigning clients. Then for each round, clients train the cluster-level model on their local data and calculate the importance weight of each aggregated cluster model $M_k$ on their local dataset via Equation 1. Here $\upsilon_{ck}$ is the normalized sum of correctly predicted data points $n_{ck}$ on local dataset $D_c$ of client $c$. The importance weights are used to generate a single-shot personalized model through the weighted aggregation of cluster-level models using Equation 2.

$$\upsilon_{ck} = n_{ck}/n_k \in [0, 1] \mid k \in [K] \tag{1}$$

---

**Algorithm 1** PI-FL (Server)

---

**Input:** $R$: Rounds, $P_r$: Pre-training rounds, $K$: Number of clusters, $M_k$: Cluster-level model of cluster $k \in K$, $M_G$: Global model at aggregator, $N$: Number of clients, $C$: Number of classes in dataset, $\zeta_a$: Available Clients, $N_p$: Number of clients to select on basis of performance, $N_r$: Number of clients to select randomly for each cluster, $\zeta_k$: Clients selected for training in cluster $k \in K$, *FedAvg*: [2], *F1-Scores*: [53], *sort()*: Python 3.7 Timsort implementation [54]

1   **for** *each round $r \in R$* **do**
2     $\zeta_k = SelectClients(r)$ for each cluster $k \in K$
3     **for** *cluster $k \in K$* **do**
4       Server sends cluster-level model $M_k$ for training to clients in $\zeta_k$
5       Token Manager collects bid payments from all willing clients via Eqn. 4
6       Token manager updates available tokens for round $r$ via Eqn. 5
7       $U_k \leftarrow$ model updates received from clients in $\zeta_k$
8       $M_k = FedAvg(U_k)$

9   **Function** SelectClients($r$)
10    **if** $r = 0$ **then**
11      **for** $k = 1$ *to* $K$ **do**
12        $\zeta_k^* \leftarrow$ Scheduler randomly assigns clients from $\zeta_a$.
13      **return** $\zeta_k^*$
14    **else if** $r > 1$ **then**
15      **for** $i = 1$ *to* $N$ **do**
16        $\theta_i \leftarrow ClientPreferences(M_1, ..., M_k) \mid \forall k \in [K]$ // from Algorithm 2
17        Server calculates marginal contributions $\psi_{ki}$ of each client within its cluster with Shapley Values approximation in Algorithm 3 $\mid \forall k \in [K], \forall i \in [N]$
18        //Profiler sorts clients on the basis of their marginal contributions and preference bids
19        $S_c = sort(\theta_i, \psi_{ki})$
20        **for** $k = 1$ *to* $K$ **do**
21          $\zeta_k^* \leftarrow N_p$ clients selected from $S_c$ and $N_r$ clients randomly from $\zeta_a$ by Scheduler.
22      **return** $\zeta_k^*$

---

$$P_{ck} = \sum_{k=1}^{K} v_{ck} \times (\omega_k) \tag{2}$$

Here the $P_{ck}$ is the personalized model of client $c$ in cluster $k$ and $\omega_k$ is the weight vector of $k$ cluster model. Using this, clients generate single-shot personalized models offline according to their dynamic data needs. The client-centric clustering and participation method enhances the appeal of pFL for clients and in doing so also provides them the opportunity to customize their personalized model offline in case their requirements which are unknown to the server change during training. Clients can also make informed decisions on participating in training clusters based on their budget and past rewards, using importance weights and knowledge of previous rounds. They convey their preferences to the aggregator by submitting bids for the cluster they wish to participate in for the next training round.

The profiler calculates the marginal contributions of each client after every round using Shapley Values approximation (Algorithm 3), aiding scheduling by providing data quality information to the scheduler. The Shapley Value approximation derivation from Appendix is used to avoid the computational expense of calculating Shapley Values for multiple clients. PI-FL also includes a mode to facilitate clients to form well-defined initial clusters. So the clients can avoid the decision-making process in the beginning and streamline their spending when the client contributions and cluster distributions are unclear. For this, the profiler and the scheduler module facilitate forming the initial clusters by training for some pre-training rounds. This is done as client contributions and similarity metrics that the clients use among other metrics to make decisions about joining clusters are initially unknown. After pre-training, the profiler calculates per-class F1-Scores $\xi$ of all client local models on an IID test dataset [53]. Then the profiler with the help of scheduler clusters clients for the next training round using the K-Means clustering [55] algorithm with the most varying F1-scores $V_{F1}$ from $C$ total classes. Equation 3 shows the calculation of $V_{F1}$ where $C$ is the number of total classes and $N$ is the number of all available clients.

$$V_{F1} = var(\xi_i) \in [1, C] \mid \forall i \in N \tag{3}$$

We perform all our evaluations for PI-FL without this feature, but this is an added feature that PI-FL includes for faster convergence and to save clients' costs. We also realize the constraints in choosing all the clients for training, which is why clients that reply within threshold time in pre-training rounds are used to calculate F1 scores. The remaining clients are considered unexplored and assigned to clusters randomly, they can later settle into appropriate clusters through preference and contribution selection.

---

**Algorithm 2** PI-FL (Client)

**Input:** $T_h$: Importance weight threshold, $K$: Number of clusters, $M_k$: Cluster-level model of cluster $k \in K$, $D$: Local dataset of client,

23 **Function** ClientPreferences($M_1, ..., M_k$)
24     **for** *each cluster $k \in K$* **do**
25         **for** *each data point $d \in D$* **do**
26             The client computes $v_k$ importance weight of $M_k$ model for each data point $d$ via Eqn. 1
27         **if** $v_k > T_h$ **then**
28             Client adds cluster $k$ to client's preference bids list $\theta_i^*$
29     The client generates personalized model $P_{ck}$ via Eqn. 2
30     **return** $\theta_i^*$

---

## 3.2 Token Manager

The token manager acts as a bank to orchestrate and keep track of transactions between different clients. At the start of each training round the token manager holds an auction for each cluster, and the clients that want to participate in that cluster place their bids using tokens. The token manager forwards the list of willing clients to the scheduler to select clients for training. It also deducts payments from the willing clients/consumers via Equation 4. Here $\tau_i$ is the tokens owned by client $i$, $\zeta_k$ are the clients willing to join cluster $k$, and $\tau_p$ in this and all following Equations is the per round bid amount to be paid by each client for participation.

$$\tau_i = \tau_i - \tau_p \mid i \in \zeta_k^* \tag{4}$$

The tokens collected as payments from clients/consumers are then added to the available pool of tokens at the Token Manager as shown in Equation 5. Here $\tau_{ar}$ are the total available pool of tokens at the Token Manager. The term $N_p$ is the number of clients selected on basis of performance and $N_r$ is the number of clients selected randomly. The significance of using $N_p$ and $N_r$ is explained in section 3.3.

$$\tau_{ar} = \tau_{ar} + (Np + N_r) \times \tau_p \mid r \in [1, R] \tag{5}$$

The token manager handles the distribution of reimbursement and rewards to each provider/client. Reimbursement penalizes degradation in the performance of providers and depends on the utility function. The utility is calculated as the percentage of average accuracy improvement of the cluster model $M_k$ over the maximum achieved accuracy in past rounds on the local data of clients in cluster $k$. The utility function is given in Equation 6 and reimbursement calculation is given in Equation 7, both metrics are calculated at the profiler which assists the token manager in reimbursement.

$$\theta = \frac{\eta \times (\gamma - \min(\gamma, \max(0.0, \frac{(Acc_{kr} - Acc_{kmax})}{Acc_{kmax}})))}{\gamma} \mid \eta \in [0, 1], \gamma \in [0, 1] \tag{6}$$

$$\tau_i = \tau_i - \tau_{ar} \times \theta \mid \theta \in [0, \gamma], \forall i \in [N], \forall r \in [1, R] \tag{7}$$

In Equation 6, $Acc_{kr}$ is the cluster-level model accuracy in the current round $r$ and $Acc_{kmax}$ is the maximum cluster-level model accuracy achieved until the current round $r$. The term $\eta$ represents the maximum portion of tokens that can be returned and $\gamma$ represents the maximum accuracy improvement that leads to the use of one full token. In Equation 7, $\tau_{ar}$ are the total number of tokens collected from consumers/clients for $r$ training round. We have used a similar approach to [28], however, they use the accuracy of the FedAvg model on an IID dataset. It is not practical to assume the presence of an IID dataset that can correspond to the data distribution of clients within a cluster which is why we rely on the local dataset of clients within that cluster to gather this information.

$$\tau_i = \tau_i + sort(\psi_{ki}, \Omega_{ki}) \times \frac{\tau_{ar}}{N_r \times \frac{(N_r+1)}{2}} \mid \forall k \in [K], \forall i \in [N], \forall r \in [R] \tag{8}$$

After reimbursement, the token manager uses the marginal contributions calculated by the profiler and sorts providers/clients by their contributions and participation record in Equation 8. Here $\psi_{ki}$ represents the marginal contributions and $\Omega_{ki}$ represents the participation records of all clients $N$ in $K$ clusters. The term $\beta$ is a normalizing term from Equation 8 in which $N_r$ are the number of providers selected for participation in round $r$. Using the ranks $\alpha$ of providers from sorting and the normalization term $\beta$, the remaining available tokens are distributed between these providers in Equation 8. Here $\tau_i$ represents the tokens owned by provider/client $i$ and $\tau_{ar}$ are the tokens available for incentive distribution at the token manager. Through reimbursements to consumers and payments to providers, the Token Manager ensures that each client receives an incentive according to their contributions in training the pFL models. By doing so, PI-FL incentivizes improvement in personalized learning, resulting in an enhancement of PMA and a decrease in opt-outs.

### 3.3 Scheduler

The scheduler selects clients for each round $r$ by the $SelectClients(r)$ function given in Algorithm 1. The scheduler receives the preference bids $\theta_i$ from the token manager, the marginal contributions $\psi_{ki}$ from the profiler for each client $i \in N$ in cluster $k \in K$, where $N$ is the total number of clients and $K$ are the total number of clusters. Using this information scheduler groups clients with similar preference bids and then sorts those clients by their marginal contributions. Then the scheduler selects $N_p$ number of clients from the sorted clients and $N_r$ number of clients randomly. Both $N_p$ and $N_r$ are tunable parameters. To reduce bias, a small portion of clients $N_r$ are selected randomly which is a technique adopted from previous works [2, 28, 56, 57]. By grouping clients with similar preferences the scheduler reduces the within-cluster bias improving the within-cluster homogeneity and a cluster model is produced that accurately represents the clients within it. Section 4 gives a theoretical analysis of how this is an important factor in improving the PMA.

## 4 Theoretical Analysis

We study the following particular case to develop insights. Suppose there are $m$ clients in total, each observing a set of independent Gaussian observations $z_{i,j} \sim \mathcal{N}(\mu_i, \sigma^2), j = 1, \ldots, n_i$, with a personalized task of estimating its unknown mean $\mu \in \mathbb{R}$. The quality of the learning result, denoted by $\hat{\mu}$, will be assessed by the mean squared error $\mathbb{E}_i(\hat{\mu} - \mu)^2$, where the expectation $\mathbb{E}_i$ is taken with respect to the distribution of client $i$.

It is conceivable that if clients' underlying parameters $\mu_i$'s are arbitrarily given, personalized FL may not boost the local learning result. To highlight the potential benefit of cluster-based modeling, we suppose that the $m$ clients can be partitioned into two subsets: one with $m_1$ clients, say $T_1 = \{1, \ldots, m_1\}$, and the other with $m_2$ clients, say $T_2 = \{m_1+1, \ldots, m\}$, whose underlying parameters are randomly generated in the following way:

$$\mu_i \sim \mathcal{N}(\beta_1, \tau^2) \mid \quad i \in T_1, \qquad \mu_i \sim \mathcal{N}(\beta_2, \tau^2) \mid \quad i \in T_2. \tag{9}$$

Here, $\beta_1$ and $\beta_2$ can be treated as the root cause of two underlying clusters. We will study how the values of sample size $n_i$, data variation $\sigma$, within-cluster similarity as quantified by $\tau$, and cross-cluster similarity as quantified by $|\beta_1 - \beta_2|$ will influence the gain of a client in personalized learning. To simplify the discussion, we will assess the learning quality (based on the mean squared error) of any particular client $i$ in the following three procedures:

**Local training**: Client $i$ only performs local learning by minimizing the local loss $L_i(\mu) = \sum_{j=1}^{n_i}(\mu - z_{i,j})^2$, and obtains $\hat{\mu}_i = n_i^{-1}\sum_{j=1}^{n_i} z_{i,j}$. Thus, the corresponding error is

$$e(\hat{\mu}_i) = \mathbb{E}_i(\hat{\mu}_i - \mu_1)^2 = \frac{\sigma^2}{n_i}. \tag{10}$$

**Federated training**: Suppose the FL converges to the global minimum of the loss, $\sum_{i=1}^{m}\frac{n_i}{n}L_i(\mu)$, $n \overset{\Delta}{=} \sum_{i=1}^{m} n_i$, which can be calculated to be $\hat{\mu}_{\text{FL}} = \sum_{i=1}^{m}\frac{n_i}{n}\hat{\mu}_i$. Consider any particular client $i$. Without loss of generality, suppose it belongs to cluster 1, namely $i \in T_1$. From the client $i$'s angle, conditional on its local $\mu_i$ and assuming a flat prior on $\beta_1$ and $\beta_2$, client $j$'s $\mu_j$ follows $\mu_j \mid \mu_i \sim \mathcal{N}(\mu_1, 2\tau^2)$ for $j \in T_1$ and $j \neq i$, and $\mu_j \mid \mu_i \sim \mathcal{N}(\mu_1 + \beta_2 - \beta_1, 2\tau^2)$ for $j \in T_2$. Then, the corresponding error is

$$e(\hat{\mu}_{\text{FL}}) = \mathbb{E}_i(\hat{\mu}_{\text{FL}} - \mu_1)^2$$

$$= \left\{\sum_{j \in T_2}\frac{n_j}{n}(\beta_2 - \beta_1)\right\}^2 + \sum_{j=1,\ldots,m, j \neq i}\left(\frac{n_j}{n}\right)^2\left(\frac{\sigma^2}{n_j} + 2\tau^2\right) + \left(\frac{n_i}{n}\right)^2\frac{\sigma^2}{n_i}. \tag{11}$$

It can be seen that compared with (10), the above FL error can be non-vanishing if $\sum_{j \in T_2} \frac{n_j}{n}(\beta_2 - \beta_1)$ is away from zero, even if sample sizes go to infinity. In other words, in the presence of a significant difference between the two clusters, the FL may not bring additional gain compared with local learning.

**Cluster-based personalized FL**: Suppose our algorithm allows both clusters to be correctly identified upon convergence. Consider any particular client $i$. Suppose it belongs to Cluster 1 and will use a weighted average of Cluster-specific models. Specifically, the Cluster 1 model will be the minimum of the loss $\sum_{j \in T_1} \frac{n_j}{n_{T1}} L_j(\mu)$, $\quad n_{T1} \triangleq \sum_{j \in T_1} n_j$, which can be calculated to be $\hat{\mu}_{T1} = \sum_{j \in T_1} \frac{n_j}{n_{T1}} \hat{\mu}_i$. By a similar argument as in the derivation of (11), we can calculate

$$e(\hat{\mu}_{T1}) = \sum_{j \in T_1, j \neq i} \left(\frac{n_j}{n_{T1}}\right)^2 \left(\frac{\sigma^2}{n_j} + 2\tau^2\right) + \left(\frac{n_i}{n_{T1}}\right)^2 \frac{\sigma^2}{n_i}. \tag{12}$$

The above value can be smaller than that in (10). To see this, let us suppose the sample sizes $n_i$'s are all equal to, say $n_0$, for simplicity. Then, we have

$$e(\hat{\mu}_{T1}) = \frac{m_1 - 1}{m_1^2}\left(\frac{\sigma^2}{n_0} + 2\tau^2\right) + \frac{1}{m^2}\frac{\sigma^2}{n_0} = \frac{m_1 - 1}{m_1^2}\left(\frac{\sigma^2}{n_0} + 2\tau^2\right) + \frac{1}{m_1^2}\frac{\sigma^2}{n_0}$$

$$= \frac{1}{m_1}\frac{\sigma^2}{n_0} + \frac{m_1 - 1}{m_1^2}2\tau^2,$$

which is smaller than (10) if and only if

$$\tau^2 < \frac{m_1 \sigma^2}{2 n_0}. \tag{13}$$

We derive the following intuitions from this analysis: **R1.** If the within-cluster bias is relatively small, the number of cluster-specific clients is large, and data noise is large, a client will have personalized gain from collaborating with others in the same cluster. **R2.** PI-FL's incentive algorithm rewards accuracy improvement reflected in PMA, which directly correlates with reducing within-cluster bias as per Equation 13. **R3.** By association, the incentive algorithm motivates clients to join similar clusters which increases cluster homogeneity and reduces the within-cluster bias. We show the impact of change in performance with an ablation study of PI-FL incentive in section 5.5.

# 5 Experimental Study

## 5.1 Experimental Setup

We use NVIDIA GeForce RTX 3070 GPUs for all our experiments. To evaluate the performance of PI-FL with other pFL models we use four datasets. A simple CNN model (32x64x64 convolutional and 3136x128 linear layer parameters) is used that can be trained on client devices with limited system resources to map Cross-Device FL settings [10] for all pFL methods.

**CIFAR10 Data.** For comparison with FedSoft [25] we use the same CIFAR10 dataset provided in their repository. This image dataset has images of dimension $32 \times 32 \times 3$ and 10 output classes. We copy different data heterogeneity conditions from [25], namely 10:90, 30:70, linear, and random. The data classes are divided into two clusters $D_A$ and $D_B$. In the **10:90** partition, 50 clients have 90% training data from $D_A$ and 10% from $D_B$, while the other 50 have 10% training data from $D_A$ and 90% from $D_B$. The **30:70** partition is similar to 10:90 except that the distribution ratios are 30% and 70%.

**EMNIST Data.** This image dataset has images of dimension 28 x 28 and 52 output classes where 26 classes are lower case letters and 26 classes are upper case letters. Same as CIFAR10 data, we use the 10:90 and 30:70 data partitions and also include linear, and random partitions. In **linear** partition, client k has $(0.5 + k)\%$ training and testing data from $D_A$ and $(99.5 - k)\%$ training data and testing data from $D_B$. In the **random** partition, client k is assigned a mixture vector generated randomly by dividing the $[0, 1]$ range into S segments with $S - 1$ points drawn from $Uniform(0, 1)$. The training and testing data are then assigned based on this vector from $D_A$ and $D_B$. Similar to [6, 30, 37], we also divide the EMNIST dataset into $K$ clusters, where $K = \frac{C_t}{C_p}$, $C_t$ are total classes and $C_p$ are the classes owned per party with no overlap of data between clusters.

**Synthetic CIFAR10.** This is a synthetic dataset created from the CIFAR10 dataset and contains the same hetrogenous partitions of 10:90, 30:70, linear, and random. The only difference is that

the training and testing data distributions are different to simulate dynamic data at the clients. For example, in **10:90** 50 clients have 90% training data with 10% testing data from $D_A$ and 10% training data with 90% testing data from $D_B$ and vice versa. Similar to this, all the other partitions also have inverse training and testing data distributions. The reason for separate training and testing data distributions are explained in further depth in Appendix.

## 5.2 Focus of Experimental Study

First, we compare the clustering ability of PI-FL with a recent clustering-based pFL algorithm [25]. Second, we show how PI-FL compares with other non-clustering pFL models with a simple test accuracy comparison. Taking it one step further, we provide a comparison of PI-FL and other clustering and non-clustering pFL models in terms of reduction in opt-outs and PMA maintenance in section 5.3. Lastly, in section 5.5 we show that including client preferences while clustering yields better personalization results because clients can make decisions based on knowledge restricted to the aggregator server.

Table 1: Test accuracy on CIFAR10

| | PI-FL | | | | FedSoft | | | |
| | 10:90 | | 30:70 | | 10:90 | | 30:70 | |
| | c0 | c1 | c0 | c1 | c0 | c1 | c0 | c1 |
|---|---|---|---|---|---|---|---|---|
| $\theta_0$ | **63.7** | 41.3 | 58.0 | 57.7 | 48.9 | 49.5 | 48.0 | 48.4 |
| $\theta_1$ | 43.7 | **63.8** | 58.6 | 58.8 | 50.7 | 49.6 | 50.0 | 50.0 |

Table 2: Test Accuracy of pFL methods on EMNIST

| Partitions | Ditto | FedProx | FedALA | PerFedAvg | FedProto | PI-FL |
|---|---|---|---|---|---|---|
| 10:90 | 85.78±4.84 | 75.15±4.81 | 75.54±4.65 | 87.5±3.79 | 71.95±1.39 | **87.5±3.66** |
| 30:70 | 75.96±4.54 | 79.74±4.01 | 78.42±3.21 | 76.63±3.94 | 59.7±4.71 | **85.07±3.36** |
| Linear | 75.3±5.08 | 82.84±2.7 | 82.04±3.61 | 80.82±3.53 | 62.63±4.93 | **83.4±4.85** |
| Random | 77.82±6.79 | 80.93±4.42 | 78.98±5.07 | 83.31±5.19 | 68.43±5.65 | **86.21±4.34** |

## 5.3 Test Accuracy performance study.

**Effectiveness of clustering.** We evaluate the performance of cluster-level models using holdout datasets sampled from the corresponding cluster distributions ($D_A$ and $D_B$). To demonstrate the effectiveness of our proposed PI-FL method, we compare it with a recent cluster-based pFL algorithm called FedSoft using CIFAR10 data. We use the same parameters as in [25], with $N = 100$ clients, batch size 128, and learning rate $\eta = 0.01$, and perform training for 300 rounds. Table 1 presents the test accuracy for the **10:90** and **30:70** partitions with PI-FL. We observe that PI-FL performs better for the 10:90 partition, where each cluster dominates one of the distributions. With PI-FL, clients that have a greater portion of data from $\theta_0$ prefer to train in cluster $c_0$, achieving 63.68% accuracy, while clients with a greater portion of data from $\theta_1$ prefer to train in cluster $c_1$, achieving 63.82% accuracy. FedSoft cluster-level models, on the other hand, achieve 50.7% and 49.6% for 10:90 data. It is worth noting that FedSoft is unable to cater to different partitions of data through its clustering mechanism, and the performance is adversely impacted by increased heterogeneity. Moreover, cluster-level models in FedSoft are unable to dominate a single distribution of data. As expected, the performance for the 30:70 partition is not as good as it is a less heterogeneous partition than the 10:90 partition. Neither cluster dominates a single distribution, and the clients with different distributions are not clearly differentiated for training with different clusters. Additionally, the cluster-level models $c_0$ and $c_1$ have similar performance with either distribution ($\theta_0$ and $\theta_1$), as FedSoft promotes personalizing models when clients have a greater percentage of shared data. This generates cluster-level models that cannot represent a single distribution and do not perform as well as PI-FL with non-IID data.

**Comparison with non-clustering pFL models.** Table 2 shows a test accuracy comparison of PI-FL with other recent pFL algorithms. This table shows that some pFL models are able to perform well for individual partitions such as Ditto for 10:90, FedProx and FedALA for Linear, and PerFedAvg for Random, however, PI-FL is able to maintain its performance for all partitions.

## 5.4 Effectiveness of PI-FL in opt-outs reduction and PMA maintenance.

Each client's natural aim is to create a model that maximizes its test accuracy. Clients can have different thresholds of how much should be the least accuracy gain for it to participate in pFL, and we define this self-defined threshold as $\rho_i$, $i \in [N]$. Since each client can have its own definition of the threshold requirement, we define $\rho_i$ as the test accuracy achieved by client $i$ if it used FedAvg. So $\text{PMA}_i$ shows the gain in performance from pFL compared to vanilla FL using FedAvg for client $i$ in $N$. PMA is similar to GMA from [37], however, creating a single global model may not be appealing for all clients as we show in section 4 and verify in section 5.4. We formally define PMA and opt-outs in Equation 14 and 15 respectively, where $f_i(w_k)$ is the test accuracy achieved by pFL.

$$\text{PMA}_i = f_i(w_k) - \rho_i \mid i \in [N], k \in [K] \tag{14}$$

$$\text{opt-outs} = \frac{1}{N} \sum_{i=1}^{N} f_i(w_k) < \rho_i \mid i \in [N], k \in [K] \tag{15}$$

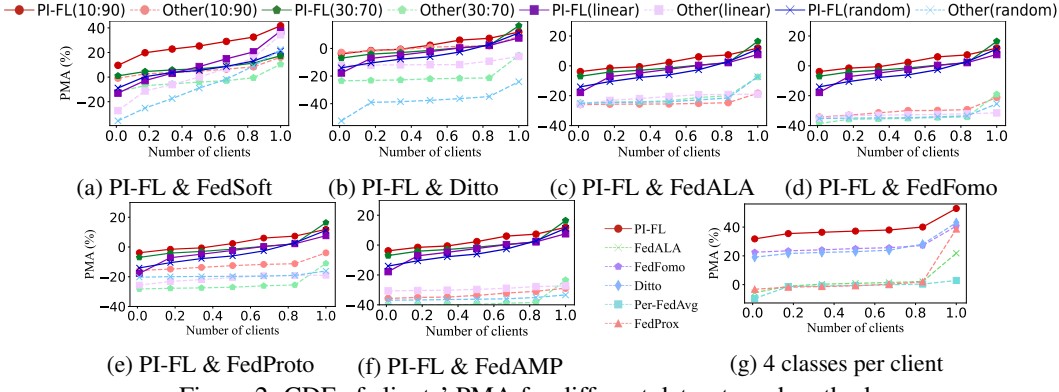

(a) PI-FL & FedSoft    (b) PI-FL & Ditto    (c) PI-FL & FedALA    (d) PI-FL & FedFomo

(e) PI-FL & FedProto    (f) PI-FL & FedAMP    (g) 4 classes per client

Figure 2: CDF of clients' PMA for different datasets and methods

Figure 2 shows the empirical Cumulative Distribution Function (CDF) plot of PMA for all clients with CIFAR10 data using FedSoft and PIFL and with EMNIST dataset for all other pFL models. PI-FL particularly outperforms for the 10:90 partition in terms of PMA as this is the most heterogeneous data partition as can be seen in Figure 2a. The EMNIST dataset is less heterogeneous as it has more classes per client compared to CIFAR10 which is why FedAvg is able to perform relatively well and there is less room for improvement with personalizing. PI-FL maintains the PMA and also improves it, particularly for the 10:90 and 30:70 partitions where other pFL solutions lack. We also test on a more heterogenous case where the dataset is divided into 52 clusters and each client owns 4 maximum classes. Figure 2g shows that while other pFL solutions perform better than FedAvg only Ditto and FedProto come relatively close to PI-FL, however, PI-FL outperforms them both by approximately 15% in terms of PMA. The FedProx, FedALA, and PerFedAvg opt-out ratios are 0.64, 0.31, and 0.68, respectively. Ditto, FedFomo, and PI-FL have no opt-outs. This goes to show that PI-FL is not only able to reduce the opt-outs but also improves the PMA under all data heterogeneity conditions.

## 5.5    Advantages of including client preferences in pFL.

We show that PI-FL can maintain the test accuracy of personalized models even in case of dynamic data at the client or a new unseen client accidentally being added to the wrong cluster. Figure 3 shows the CDF of clients' personalized model test accuracy after training for 500 rounds. PI-FL is robust to variations in clients' local data, while FedSoft is less effective due to its clustering approach being based on the server's perspective, which lacks access to clients' private data and limits its ability to make accurate clustering decisions.

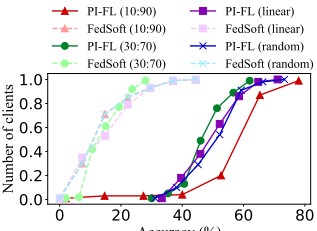

Figure 3: PI-FL and FedSoft with Synthetic CIFAR10 data

**Ablatian study with Incentive in PI-FL.** To measure the impact of incentive provision on personalized model generation we test PI-FL with incentives enabled and disabled. Figure 4 shows the CDF of clients' personalized model test accuracy with the Synthetic CIFAR10 dataset. Except for the 30:70 partition, the accuracy for all other partitions is higher with the incentive enabled. We argue that the test accuracy for 30:70 is low in this case because it is a less heterogeneous data case and PI-FL performs best in cases where data is highly heterogeneous and requires personalized learning. Further details of the experimental setup and impact of incentive on clustering are discussed in the Appendix.

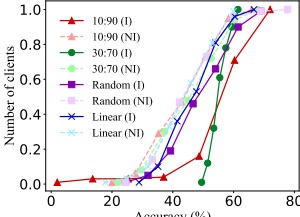

Figure 4: PI-FL with and without incentive (I/NI)

## 6    Conclusion

In this paper, we proposed PI-FL to address the challenges of incentive provision in pFL for increasing consistent participation by providing appealing personalized models to clients. PI-FL client-centric clustering approach ensures accurate clustering and improved performance even in case of dynamic data distribution shift of the client's local data or inadvertently mistaken clustering decision by the client. Unlike prior works that consider incentivizing and personalization as separate problems, PI-FL solves them as interrelated challenges yielding improvement in pFL performance. Extensive empirical evaluation shows its promising performance compared to other state-of-the-art works.

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
