# OpenReview forum: "PI-FL: Personalized and Incentivized Federated Learning"
_NeurIPS.cc/2023/Conference — Submitted to NeurIPS 2023_

### Official Review · Reviewer_H8pJ · 2023-07-03

**Soundness:** 3 good
**Presentation:** 3 good
**Contribution:** 3 good
**Rating:** 7
**Confidence:** 4

**Summary:**

This paper proposes and analyzes a mechanism for incentivizing data contributions to federated learning systems in a context where contributors consider both monetary incentives for data contributions and the ability to experience the benefits of personalized systems.

The paper provides background on incentivized federated learning, describing the problem setting and proposed mechanisms, and theoretically analyzes a particular subset of this problem category to understand impacts of how underlying data that’s being modeled is distributed. Then, the authors perform experiments with classification datasets, with the assumption that classification datasets that might be under the control of a single entity in a classical setting are distributed between 100 clients and all data points belong to one of two clusters.


**Strengths:**

The paper has a number of strengths:

The paper begins with a very helpful overview of work in federated learning. The paper will be of interest beyond the federated learning-focused sub-community. The current draft provides strong motivation by explaining the large gap in incentivized federated learning. It seems with further development this approach could be impactful in practice.

The paper combines theoretical analysis of a new mechanism (which has the key insight of accounting for both monetary incentives and personalization-as-an-incentive) with learning experiments. This combination can help convince readers the proposed approach is useful. While there is some room to provide more details about the intended "data contributor scenario" and "data user scenario" (see below), overall I think the methods lean more towards being an overall strength vs. a weakness.

Presentation of results was strong overall.


**Weaknesses:**


While readers will appreciate the combination of theoretical analysis and experiment, it was not entirely clear exactly how the specific research questions being answered in each section were chosen (i.e., the “intuitions” R1-R3 seem useful but were not well prefaced). The experiments could be better motivated (more below in "Questions").

The paper is pretty abstracted away from any specific platform or use case where data contributors and data users interface via a particular FL architecture.

There may be an opportunity to strengthen the paper by specifying which ML use cases / contexts map well to the needed assumptions. In particular, while the paper relies on the existence of prior work on FL to motivate the need for FL, it’s not clear how many platforms running actual incentive systems (e.g. data markets, markets for crowdwork, etc.) are also supporting FL.

In general, I worry this paper may undersell its contribution by not providing enough details about what an on-the-ground implementation of the system would look like. Of course, this same criticism can be applied to a large body of similar work (including the RW here) and so it's not existential. Given the specific intended contribution here of proposing a new way to think about incentives in PFL, emphasizing plausibility may be especially valuable.


**Questions:**

-	A core motivating claim of the paper is that ML requires an explicit incentive system for data collection. I personally think this a good aspirational goal for the community, but was skeptical when reading that this is an accurate claim, given major research attention being paid to models that primarily focus on scraped data (that doesn’t really map to traditional notions of consent or incentives). This doesn’t invalidate any claims here, but I think is important to discuss and acknowledge. For instance, see the claims in line 30 of the current draft.
-	Outside of the experiments, the current draft is not specific about any particular use case or specific kind of data generation scenario. For instance, in the paragraph starting on line 50, I had a decent sense of the abstract ideas being presented but had a lot of trouble imagine the specific scenarios faced by data contributing users. What specific kind of data are they imagining being incentivized to share? To me, this feels like a very valuable question to at least partially address given the core contribution of the paper is about incentives facing people.
-	There are opportunities to further discuss the gap between modeling personalization in terms of traditional supervise classification with clusters versus something like a recommender system. As a reader, I was hoping to see some mention of this, as the conceptual contribution of the paper that personalization should be consider part of the incentive mechanism seems especially relevant to actual recommendation tasks.


**Limitations:**

There are opportunities for the paper to be a bit more upfront about the limitations of focusing on incentive systems without discussing any specific populations of data contributors or specific use cases (beyond computer vision evaluation). While this paper seems to be part of a longer conversation between research papers that is primarily mathematical and leans heavily on assumptions about rational agents with programmatic approaches to computing utility, given that a key goal of having practical advantages mentioned in Related Work, addressing some practical use cases could make the work more convincing.

---

> ### Author Rebuttal · Authors · 2023-08-08
>
> **Answer 1.**
>
> We deeply appreciate the reviewer's insightful questions, and we would like to address them in a consolidated response.
>
> In the rapidly evolving landscape of AI, several AI marketplaces such as ModelPlac [1], GravityAI [2], AWS AI Marketplace [3], and AI Marketplace [4] are emerging that cater to both model and data needs. However, it's important to highlight a significant distinction. While these platforms do offer data and model training services, the data they provide is often static and not as current as the data generated freshly on individual client devices. Moreover, there are considerable challenges associated with transporting data, including high costs and privacy concerns.
>
> For applications centered around recommendations and user-driven data, the inclusion of the most recent user trends is of paramount importance. Federated learning with personalized models for users presents an effective approach to achieving this objective. The existing market for Federated models remains limited due to the dearth of incentives compelling clients or users to contribute their valuable, up-to-date data that can contribute to the creation of high-quality models.
>
> Our work is strategically designed to address this gap and accelerate advancements in the creation of architectures and frameworks that facilitate the growth of this domain. By emphasizing the significance of fresh and relevant data from client devices, we aim to establish a foundation for incentivizing active user participation and ultimately producing models of superior quality. This endeavor is driven by the understanding that incentivizing clients/users to contribute their data is pivotal to unlocking the full potential of federated learning in dynamic and personalized applications.
>
> [1] https://modelplace.ai/
> [2] https://www.gravity-ai.com/
> [3] https://aws.amazon.com/marketplace/solutions/machine-learning
> [4] https://aimarketplace.co/
>
> **Answer 2.**
>
> We extend our appreciation to the reviewer for sharing these insightful comments. It's worth noting that the concept of recommendation falls under the broader umbrella of personalization. In essence, personalization encapsulates a broader range of scenarios beyond classification, including areas such as text generation, recommendation systems, e-commerce, healthcare, advertising, education, financial services, news delivery, social media curation, travel and hospitality, entertainment experiences, and even the customization of smart devices. While we didn't explicitly delve into the distinction between personalization through clusters and personalized recommendation systems in this particular paper, we appreciate the reviewer’s valuable insights. This area does indeed warrant attention and is an avenue that could be explored in future research to comprehensively address the nuanced relationship between these various forms of personalization.
>
> We will include this discussion in the final draft of the paper.

---

> > ### Comment · Reviewer_H8pJ · 2023-08-15
> > **Quick response to answers**
> >
> > Thanks to the authors for this additional info!
> >
> > Regarding answer 1 -- this totally makes sense. I do think readers will, in general, understand the value and impact of model/data markets. I think this line of motivation is definitely compelling.
> >
> > Just to clarify my original comment, I meant to say that a large portion of data records in certain contexts currently are not being collected via a functional market mechanism (or one could argue there is a sort of market, just with intense information asymmetry). To provide a few examples of what I mean, I'm thinking of scenarios in which ML/AI systems are trained using data collected via surveillance-style instruments, or via log data that is highly obfuscated.
> >
> > I don't think this necessarily contrasts with anything in your rebuttal comment. My original comment was motivated by the fact I felt the original paper draft didn't really talk about this tension at all (which is fair given scope)
> >
> > Regarding answer 2 - this seems totally reasonable to me!

---

### Official Review · Reviewer_bwzk · 2023-07-04

**Soundness:** 2 fair
**Presentation:** 2 fair
**Contribution:** 3 good
**Rating:** 5
**Confidence:** 4

**Summary:**

This paper proposed PI-FL, an incentive-aware federated learning method that produces clustered personalization of client models. This work uniquely considers the client contribution assessment in the multi-objective setting of clustered personalization. The clients are also given the freedom to choose the cluster based on their incentives. Therefore, this paper could be valuable in exploring the topic of incentivization in the very specialized area of clustered personalization. Rather extensive empirical experiments were also carried out by the authors to demonstrate the practicability and effectiveness of PI-FL.

**Strengths:**

1. Recognition that personalization and incentivization in FL should be considered as interrelated challenges. In fact, incentivization is easier to be achieved when client models are allowed to be personalized.
2. In addition to the proposed method, the paper provides good theoretical analyses of it through toy examples that are easy to understand. The analysis also offers good and interpretable insights.
3. The types of empirical experiments carried out are rather extensive and demonstrate the effectiveness of the method. However, the comparison to baselines and the dataset choices can be further improved to be more comprehensive (sometimes omitting certain settings for certain methods).

**Weaknesses:**

1. This paper might need more concise and precise writing in Section 3 Proposed Methodology. The description of the methodology is long-winded and many design choices are not backed by theoretical justifications.
2. The theoretical analysis focuses on the gain from collaboration but neglects the theoretical aspects of the mechanism that incentivizes participation and honest data contribution.

**Questions:**

**Major**

1. The role of tokens in the incentive mechanism is unclear in the writing. Do all the clients start with the same number of tokens? Do they pay for the tokens after use? If tokens are involved in the incentivization, I do not see descriptions of how they interplay with model rewards (e.g., PMA). For example, should we ask for more money from an agent that achieves more PMA? Is it still fair when some agent uses more tokens than others? Also, how do you design the token-to-contribution denomination? Please help to clarify.
2. Collaborative fairness is included as a major contribution of the proposed method. This notion is undefined in the paper itself. One possible place to find reference is [1], which is defined to be a high correlation between the performance of models received by the clients and the standalone performance of the client's local data. Can you produce a result that compares the collaborative fairness of various methods?
3. The scenarios that the proposed method work seem to be too specialized and extreme. For example in Figure 4, PI-FL does not work so well in the 30:70 partition. It seems that a very extreme heterogeneity is needed to see the superiority of the proposed method, how to then tackle the case when some agents have very heterogeneous data while the rest of the agents have rather balanced datasets?
4. The empirical results in Section 5.3 is not very convincing given the relatively easy dataset and limited baselines used. How about FedGroup and [29], are they being compared anywhere in terms of the clustering effectiveness?
5. Does Table 1 and 2 show average accuracy? How many clients are there? Is the number of clients not so important in this experiment?
6. Why is CIFAR-10 only tested on FedSoft and CIFAR-10? Can you provide a more complete comparison in the Appendix?

**Minor**

7. Some Line references for the Algorithms in Section 3 first paragraph seem to be incorrect.
8. Should you add an indicator function in Equation (15)?

**Limitations:**

1. It might be necessary to discuss the incentive and fairness more clearly. For example, we can see from Figure 2 that clients have varying PMA, which indicates different levels of benefits were gained by different clients in the collaboration. How should this phenomenon be viewed under the lens of fairness? Also, the opt-outs in a very coarse binary indicator and might not be a good way to see whether clients are really incentivized.

---

> ### Author Rebuttal · Authors · 2023-08-08
>
> **Answer 1.**
>
> We appreciate the reviewers for raising this important issue. In our future work, we intend to dive deeper into this topic, Specifically, we will explore variable pricing methods. In this paper, we have followed other state of art token-based incentive federated learning frameworks [1-3] that assume all clients commence with the same number of tokens.
> Increasing the Personalized Model Accuracy (PMA) motivates clients to join precise clusters. However, if we were to request extra money from high PMA devices as the reviewer suggests, it would actually work against our goal. This approach will discourage clients from striving for higher PMA because of the increased costs involved. This, in turn, will lead to less accurate clustering decisions as clients' motivation to improve their PMA decreases resulting in suboptimal outcomes.
>
> [1] Jingoo Han et al. (2022). "Tokenized Incentive for Federated Learning." In Proceedings of AAAI International Workshop on Trustable, Verifiable and Auditable Federated Learning (FL-AAAI 22)
>
> [2] Shashi Raj Pandey et al. (2022). "FedToken: Tokenized Incentives for Data Contribution in Federated Learning." Workshop on Federated Learning: Recent Advances and New Challenges, in Conjunction with NeurIPS 2022 (FL-NeurIPS'22).
>
> [3] U. Majeed et al. (2023), "FL-Incentivizer: FL-NFT and FL-Tokens for Federated Learning Model Trading and Training," in IEEE Access, vol. 11, pp. 4381-4399, 2023, doi: 10.1109/ACCESS.2023.3235484.
>
> **Answer 2.**
>
> Existing metrics of collaborative fairness cannot be used to calculate fairness in pFL as existing solutions of pFL do not include incentivization which is needed to ensure collaborative fairness. Thus, calculating collaborative fairness first requires incorporating incentivization in pFL and then proposing new metrics to measure it. Our work is the first step in this direction where we propose a new metric called PMA which calculates/measures the gains of each client with personalized learning. In addition, we incorporate another metric called opt-out which along with PMA which ensures collaborative fairness per client. If the PMA representing the gains of a client falls below a certain threshold collaboration fairness is ensured by that client opting out of future training increasing the number of opt-outs, so clients' participation is directly related to their gains ensuring fairness in collaboration. Using this metric, comparative results in Table 7 of the paper show that PI-FL reduces opt-outs/increases fairness compared to various other pFL methods.
>
> **Answer 3.**
>
> In the specific scenario outlined by the reviewer, a segment of clients possesses balanced datasets, while another group holds highly heterogeneous data. Within this context, the subset of clients with balanced datasets can make meaningful contributions across multiple clusters, given their possession of all relevant labels. Thus, it is evident that PI-FL is well-equipped to effectively manage this specific scenario as well.
>
> **Answer 4.**
>
> In this paper, we assess all algorithms using the same datasets as the ones used in the original algorithm papers. We do this deliberately to play to the other algorithm’s strengths instead of biasing the evaluation in favor of our own approach. Moreover, to address the reviewer’s concerns we extended our analysis to include two new algorithm comparisons namely IFCA [1] and FedEM [2], using the CIFAR10 and Synthetic CIFAR10 benchmark datasets from FedSoft under the same conditions detailed in Section 5.3. These results are shown in Table 1 of the uploaded file.
>
> [1] A. Ghosh et al. (2022). "An Efficient Framework for Clustered Federated Learning." IEEE Transactions on Information Theory, 68(12), 8076-8091. doi: 10.1109/TIT.2022.3192506.
>
> [2] Othmane MARFOQ et al. (2021). "Federated Multi-Task Learning under a Mixture of Distributions." 35th Conference on Neural Information Processing Systems (NeurIPS 2021).
>
> **Answers 5 and 6.**
>
> Yes, the inclusion of the number of clients in Table 1 is unnecessary, as this table presents the cluster-level model accuracies on holdout datasets, rather than individual clients' personalized accuracies. This mode of evaluation aligns with similar practices, as in FedSoft [5]. It's important to note that the specific number of clients for Table 2 is explicitly provided in Section 5.3 of the paper (N=100 clients), where a comprehensive explanation of both tables is offered.
>
> We would like to point out that all algorithms have been evaluated on the datasets used in the original papers. In addition, to address the concern of more evaluations with the CIFAR10 dataset we would like to refer the reviewer to Figure 7 in section A.4, where a comprehensive comparison among four distinct algorithms—namely FedFomo, FedProx, FedALA, and Ditto—is presented. This comparison is on a challenging dataset, specifically the CIFAR10 dataset with only two classes per client. In addition, the newly uploaded results with FedEM and IFCA also use the same CIFAR10 dataset as in FedSoft.
>
> [5] Yichen Ruan et al. (2022). "FedSoft: Soft Clustered Federated Learning with Proximal Local Updating." The Thirty-Sixth AAAI Conference on Artificial Intelligence (AAAI-22)
>
> **Answers 7 and 8.**
>
> We will duly address the reviewer's suggestions and make the necessary adjustments to the references and equations accordingly.

---

> > ### Comment · Reviewer_bwzk · 2023-08-13
> > **Thanks for the rebuttal**
> >
> > 1. I am still confused: Do the clients pay for the tokens they use? Is there any cost involved for clients to obtain a better model? My original thinking was that if you reward a client more (with a better PMA model) then the client has to contribute more (maybe tokens?) to deserve the good reward right?
> >
> > This is in fact linked to the Limitation I have pointed out in the review. Incentivization is inevitably linked to fairness issues and needs to be discussed.
> >
> > 2. Sorry for leaving a [1] there in the original review and forgetting to paste the reference there. I just realized the oversight. It should be “[1] Gradient-driven rewards to guarantee fairness in collaborative machine learning. NeurIPS, 2021.” As mentioned in Limitations, opt-out is a very coarse binary indicator and might not be good enough to compare levels of incentivization.
> >
> > 3. I do not explicitly restrict my scenario to label splits: The highly heterogeneous data could contain all labels as well, just following a Dirichlet $Dir(\alpha)$ with a smaller concentration parameter $\alpha$. Going back to the question, I might want an explanation for the bad performance of the 30:70 partition. Will a 40:60 partition be even worse? Do you want to try an experiment?

---

> > > ### Author Response · Authors · 2023-08-15
> > > **Further clarification of concerns**
> > >
> > > **Comment 1. I am still confused: Do the clients pay for the tokens they use? Is there any cost involved for clients to obtain a better model? My original thinking was that if you reward a client more (with a better PMA model) then the client has to contribute more (maybe tokens?) to deserve the good reward right?
> > > This is in fact linked to the Limitation I have pointed out in the review. Incentivization is inevitably linked to fairness issues and needs to be discussed.
> > > Sorry for leaving a [1] there in the original review and forgetting to paste the reference there. I just realized the oversight. It should be “[1] Gradient-driven rewards to guarantee fairness in collaborative machine learning. NeurIPS, 2021.” As mentioned in Limitations, opt-out is a very coarse binary indicator and might not be good enough to compare levels of incentivization.**
> > >
> > > **Response:** We appreciate the reviewer’s keen insights and added reference. For clarification, we would like to explain the working and purpose of PI-FL’s incentive system design.
> > >
> > > **Consumer and Provider Roles:** As explained in line 68 of the paper “**The incentive mechanism in PI-FL treats clients as both providers and consumers.** As a consumer, the client tries to attain a certain level of personalized model appeal, so it pays the provider to spend resources to participate in training for the said model in each round. Whereas as a provider, the client earns a profit based on its marginal contribution to training the cluster models.”
> > >
> > > **Fairness for consumers:** Yes, the clients with consumer’s profile pay for the tokens they use. To ensure that the consumers only pay for what they are getting they have the option of opt-out, where consumer clients can opt-out if they are not getting a model with a PMA over the desired threshold. **To overcome the coarse-grained limitation of opt-out**, the consumers also get a partial reimbursement of their tokens spent (Section 3.2: Equations 6 and 7) if the resulting PMA was not upto their expected threshold.
> > >
> > > **Opportunity fairness:** In this paper, we have followed other state of art token-based incentive federated learning frameworks that ensure no consumer or a sub-group of consumers is able to manipulate the overall training process by spending more tokens which is why we have a single pricing model for all consumers. Starting with the same number of tokens also ensures equal opportunity for all consumer clients.
> > >
> > > **Fairness for providers:** We also ensure collaboration fairness among clients with the provider’s profile. A provider that has more marginal contribution (calculated using Shapley Values) in training the model receives more token rewards according to Algorithm 1. The second incentive for providers similar to consumers is increased PMA. Both rewards are dependent on accurate clustering choices from individual provider clients. Hence, to achieve these incentives each client makes better clustering choices which lead to clusters that have similar client objectives (good personalization) and this increases the provider client’s incentives (PMA and tokens). This is theoretically supported in Section 4.
> > >
> > >
> > >
> > >
> > > **Comment2: I do not explicitly restrict my scenario to label splits: The highly heterogeneous data could contain all labels as well, just following a Dirichlet
> > >  with a smaller concentration parameter
> > > . Going back to the question, I might want an explanation for the bad performance of the 30:70 partition. Will a 40:60 partition be even worse? Do you want to try an experiment?**
> > >
> > > **Response:** Here we would like to emphasize the point that personalization is a solution for increased data heterogeneity. With IID data personalization loses its purpose as that can be solved by simple FedAvg. Here 50:50 would be a completely IID case and as the reviewer suggested 40:60 is relatively slightly heterogeneous. For 50:50 PI-FL will have a similar performance as FedAvg and with 40:60 PI-FL will perform slightly better. We want to assure the reviewer that we can share the results of the 40:60 evaluation before the discussion period ends.
> > >
> > > We thank the reviewer for the insightful remarks and have tried our best to answer each concern diligently.

---

> > > ### Author Response · Authors · 2023-08-17
> > > **Experiment results for 40:60**
> > >
> > > **Comment2: I do not explicitly restrict my scenario to label splits: The highly heterogeneous data could contain all labels as well, just following a Dirichlet with a smaller concentration parameter . Going back to the question, I might want an explanation for the bad performance of the 30:70 partition. Will a 40:60 partition be even worse? Do you want to try an experiment?**
> > >
> > > **Response:** We have expanded our evaluation as per the reviewer's request. In addition to the analyses presented in Sections 5.3 and 5.4, we now include an evaluation of the 40:60 partition using the EMNIST dataset, following the parameters outlined in Section 5.3. Notably, in this new partition setting, our proposed PI-FL algorithm performs better compared to other algorithms under consideration. As mentioned in the prior response 40:60 is slightly heterogeneous compared to the complete IID case of 50:50 which is why we see an improved performance with personalization particularly with PI-FL.
> > >
> > >
> > >
> > > | pFL method          | FedAvg   | Ditto   | FedProx  | FedALA   | FedFomo  | FedProto | PerfFedAvg| PI-FL  |
> > > |---------------------|----------|---------|----------|----------|----------|----------|-----------|--------|
> > > | Personalized Accuracy | 76.22±0.47 | 77.93±1.1 | 82.2±0.47 | 74.65±3.51 | 79.14±2.36 | 47.65±14 | 83.3±2.94 | **85.11±2.1** |
> > > | Optouts             | -        | 0       | 26      | 0        | 100      | 0        | 0         | **0**      |
> > > | Average PMA         | -        | 1.7     | 5.9     | 0.14     | 2.9      | -28.6    | 7.07      | **8.86**   |

---

> > > > ### Comment · Reviewer_bwzk · 2023-08-21
> > > >
> > > > As mentioned, incentives are closely linked to fairness but the authors have included incentives (i.e., getting a better model than local training) in the paper without a careful and rigorous discussion on fairness (i.e., clients with higher contribution in terms of data or model gets a better reward). That is why I suggested collaborative fairness measures instead of using the coarse indicator of “opt-out”, which essentially only measures “individual rationality”.
> > > >
> > > > I would like to thank the reviewers for the additional experiment. If the method works for 40:60 less heterogeneous setting, I suppose it would work for 30:70, too. The authors could investigate and update the 30:70 experiment in the revised paper.
> > > >
> > > > Overall, I would like to keep my score. Thank you authors for your effort!

---

### Official Review · Reviewer_MX3C · 2023-07-06

**Soundness:** 3 good
**Presentation:** 3 good
**Contribution:** 3 good
**Rating:** 4
**Confidence:** 2

**Summary:**

In this paper, the authors propose a novel clustering-based pFL combined with a token-based incentive mechanism to address incentive provision in pFL. Specifically, the proposed method clusters clients based on their cluster preference (the distribution similarity), which maximizes their contribution to the clusters.

**Strengths:**

1. The proposed approach of incentive-driven clustering based pFL is novel.

2. Theoretical analysis of the proposed method is provided.

3. Experimental results on multiple datasets demonstrate promising outcomes.

**Weaknesses:**


1. Some notations are not explained:


	What's $\zeta^{*}_k$ in Eqn.(4)? Only $\zeta_k$ is explained in line 177 of page 5

	What's $\theta_i$ in line 211 of page 6? Is it the same as the $\theta$ in Eqn.(6)?


2. Some parts of the method is not explained well


	What's the difference between reimbursement and payment?

	What does the term of "token" actually means in pFL?

	What's the bidding strategy of clients?

3. Experiment:

	In pFL, the data are often heterogeneous in label shifting or feature shifting. However, the datasets used in the paper seems only related to dataset size imbalance, which are not the representative datasets used in pFL research.

	Hope to see the experimental results on CIFAR10 data with Dirichlet splitting (referring the settings in [1]).

	[1] Marfoq, Othmane, et al. "Federated multi-task learning under a mixture of distributions." Advances in Neural Information Processing Systems 34 (2021): 15434-15447.

Minor:

	Typos: In figure 2, the x-axis is the number of clients which supposed to be a positive int type, but in the figure, it ranges from 0 to 1.

**Questions:**

1. Refer weakness 1 and weakness 2.

---

> ### Author Rebuttal · Authors · 2023-08-08
>
> Ck* is the same as Ck, and theta_i originates explicitly from algorithm 2 in line 30. Theta represents the utility of the client, as defined in equation 6. To eliminate any possible confusion, we will promptly modify the symbols and rectify Ck* without delay.
>
> Furthermore, for the readers who may not be well-versed in AI incentive systems, we have provided comprehensive explanations of these terms ("payments" "reimbursement" and "tokens") in this paper. As an example, in lines 173-178, we clearly explain the token manager's role in working with the scheduler for client selection and deducting payments from participating clients based on Equation 4. Similarly, in lines 184-186 we give the definitions of reimbursement “Reimbursement penalizes degradation in the performance of providers and depends on the utility function. The utility is calculated as the percentage of average accuracy improvement of the cluster model Mk over the maximum achieved accuracy in past rounds on the local data of clients in cluster k.” For adding further clarity, the term "payments" pertains strictly to clients' contributions for participating in a particular tier to acquire the trained model. Similarly, "tokens" refers to tokens/chips/virtual currency commonly used as a form of payment in token-based incentive algorithms, as explicitly explained in the provided references. Therefore, we reiterate that in this paper, we have exhaustively explained the pivotal role of each term in our algorithm. Its also important to note that tokens are not derived from pFL, as erroneously presumed by the reviewer.  Furthermore, the bidding strategy employed is entirely consistent with other state-of-the-art token-based incentive algorithms, as aptly referenced. For a thorough understanding of these fundamental terms "payments" "reimbursement" and "tokens", we direct interested readers to the referenced sources from the PI-FL paper which we also cite here for convenience [1-6].
>
> As for the reviewer's third question, we categorically clarify that we use representative non-IID heterogeneous benchmark datasets from other state-of-the-art pFL works, as explicitly indicated in the provided references. Specifically, we implement "label-shifting" and not dataset size imbalance, as mistakenly interpreted by the reviewer. This fact is stated in section 5.1, where we mention: "We adopt different data heterogeneity conditions from [25] (FedSoft), namely 10:90, 30:70, linear, and random. The data classes are divided into two clusters DA and DB." As also acknowledged by Reviewer bwzk we have used highly heterogeneous and non-IID representative datasets.
>
> Furthermore, it's important to emphasize that the reviewer's suggestion regarding feature-shifting holds relevance primarily for Vertical Federated Learning or Hybrid Federated Learning. However, it's worth noting that this suggestion is inapplicable for Horizontal Federated Learning, exemplified by our PI-FL approach.
>
> [1] Zhang et al., "A review of incentive mechanism in peer-to-peer systems." In 2009 First International Conference on Advances in P2P Systems, pp. 45-50. IEEE, 2009.
>
> [2] Aslani et al., "A token-based incentive mechanism for video streaming applications in peer-to-peer networks." Multimedia Tools and Applications 77 (2018): 14625-14653.
>
> [3] J. Han et al., "TIFF: Tokenized Incentive for Federated Learning," 2022 IEEE 15th International Conference on Cloud Computing (CLOUD), Barcelona, Spain, 2022, pp. 407-416, doi: 10.1109/CLOUD55607.2022.00064.
>
> [4] Jingoo Han et al. (2022). "Tokenized Incentive for Federated Learning." In Proceedings of AAAI International Workshop on Trustable, Verifiable and Auditable Federated Learning (FL-AAAI 22)
>
> [5] Shashi Raj Pandey et al. (2022). "FedToken: Tokenized Incentives for Data Contribution in Federated Learning." Workshop on Federated Learning: Recent Advances and New Challenges, in Conjunction with NeurIPS 2022 (FL-NeurIPS'22).
>
> [6] U. Majeed et al. (2023), "FL-Incentivizer: FL-NFT and FL-Tokens for Federated Learning Model Trading and Training," in IEEE Access, vol. 11, pp. 4381-4399, 2023, doi: 10.1109/ACCESS.2023.3235484.

---

> > ### Comment · Reviewer_MX3C · 2023-08-15
> >
> > Thanks for your response!
> >
> > I am still confused about the experiment.  The authors mentioned in the response that "Furthermore, it's important to emphasize that the reviewer's suggestion regarding feature-shifting holds relevance primarily for Vertical Federated Learning or Hybrid Federated Learning. However, it's worth noting that this suggestion is inapplicable for Horizontal Federated Learning, exemplified by our PI-FL approach."
> >
> > Actually, feature shifting is common in pFL (HFL setting), there are a series of works aims to solve the problem of feature shift. One representative and classic work is FedBN (https://arxiv.org/pdf/2102.07623.pdf). Therefore, without further experiments, it's hard to convince me the effectiveness of the proposed work.

---

> > > ### Author Response · Authors · 2023-08-16
> > > **Thank you for the response**
> > >
> > > We are grateful to the reviewer for providing additional context about the significance of feature shifting within the framework of Horizontal Federated Learning (HFL), along with the reference to the FedBN work. We highly value your insights and regret any misunderstanding regarding the specific evaluation you sought.
> > >
> > > We acknowledge the importance of providing the required results to address the concerns effectively. Please rest assured that we will share the necessary experiment results with feature shifting before the discussion deadline.

---

> > > ### Author Response · Authors · 2023-08-18
> > >
> > > | pFL method          | FedAvg    | Ditto     | FedProx   | FedALA    | FedFomo   | FedProto   | PerfFedAvg | PI-FL     |
> > > |-----------------------------|-----------|-----------|-----------|-----------|-----------|------------|------------|-----------|
> > > | **Personalized Accuracy** | 75.74±3.41 | 83.79±2.45 | 78.81±13.6 | 83.3±2.64  | 63.91±9.91 | 46.58±20.61 | 83.77±2.52 | **84.37±2.3** |
> > > | **Optouts**             | -         | 0         | 23        | 0         | 100       | 0          | 0          | **0**         |
> > > | **Average PMA**         | -         | 7.63      | 3.07      | 7.2       | -17.06    | -48.66     | 7.77       | **8.6**       |
> > >
> > > In response to the insightful comments from the reviewer, we have extended our evaluation, as discussed in Sections 5.3 and 5.4, while retaining the same experimental conditions. In these sections, we explore non-IID data scenarios using a feature shift technique. Specifically, we construct dataset DA comprising normal images from the EMNIST dataset, and dataset DB containing images rotated by 90 degrees. This methodology for feature-shift evaluation is adopted from prior works on personalized Federated Learning [1,2].
> > >
> > > These results show that both PerFedAvg and Ditto, as referenced in the PI-FL paper, showcase good performance, closely approaching the efficacy of PI-FL. Nevertheless, it is worth highlighting that PI-FL surpasses them, not only with feature-shift non-IID data but also across a spectrum of diverse non-IID datasets detailed in Sections 5.3 and 5.4.
> > >
> > >
> > > [1] Ruan, Y., & Joe-Wong, C. (2021). FedSoft: Soft Clustered Federated Learning with Proximal Local Updating. AAAI Conference on Artificial Intelligence.
> > > [2] A. Ghosh, J. Chung, D. Yin, and K. Ramchandran. "An Efficient Framework for Clustered Federated Learning." In Proceedings of the 34th Conference on Neural Information Processing Systems (NeurIPS 2020), Vancouver, Canada.

---

### Official Review · Reviewer_mhQ3 · 2023-07-06

**Soundness:** 2 fair
**Presentation:** 3 good
**Contribution:** 3 good
**Rating:** 4
**Confidence:** 4

**Summary:**

This paper presents PI-FL, a new cluster-based personalized Federation Learning (pFL) approach. The new idea is the first to propose to consider motivation and personalization as interrelated challenges and address them through incentive mechanisms that promote personalized learning. PI-FL let clients provide incentive-driven preferences for joining clusters based on their own data distribution, and this client-centric clustering approach ensures accurate clustering and improved performance. This approach results in improved personalized model appeal (PMA) and reduced opt-outs, which in turn improves the accuracy of the clustering model. Theoretical analysis initially shows the effectiveness of the motivating algorithm. Experiments on several datasets show that the algorithm can obtain higher accuracy than the baseline.

**Strengths:**

1. This paper proposes to consider motivation and personalization as interrelated challenges and to address them through incentives that promote personalized learning. This is a good and reasonably innovative point that is informative for subsequent research on personalized federal learning.
2. The ability of PI-FL, a customer-centric clustering approach, to access the original customer data ensures accurate clustering and improved performance even in the case of dynamic distribution shifts in the customer's local data or incorrect clustering decisions by the customer, which improves the robustness of federation learning.
3. This paper is basically written very clearly. The three main modules of PI-FL: the profiler, the token manager, and the scheduler are very cleverly designed and their functions and working principles are well described.
4. The theoretical analysis is carried out in the paper to prove the effectiveness of the incentive algorithm, and the theoretical part is relatively complete.
5. The experimental comparison of test accuracy in multiple partitions shows that PI-FL can maintain good performance in all partitions. The comparison experiments of the two metrics, PMA and opt-outs, show that PI-FL can not only reduce opt-outs but also improve PMA in all data heterogeneity conditions.

**Weaknesses:**

1. The paper seems to lack the analysis about the convergence of PI-FL and the comparison with other pFL algorithms about the convergence speed.

~~2. In order to show the effectiveness of PI-FL method, the comparison between PI-FL and other clustering-based pFL algorithms about computation and communication cost should be added, and there should be more comparisons between several clustering-based pFL algorithms instead of only comparing with FedSoft.~~

3. All experiments are conducted in a simple CNN model, and the empirical study would be stronger if the authors could add some linguistic tasks, such as BERT pre-training/ fine-tuning experiments
4. Figure 1, as the overall design of PI-FL, is not drawn in enough detail, e.g., the icons representing customers are not labeled as customers. the fold lines in Figure 4 are dense and affect the perception, e.g., it is difficult to see clearly the fold lines representing 10:90 (NI).
5. The N_p parameter of the scheduler in PI-FL is the number of customers selected based on performance, and N_r is the number of customers selected randomly. N_p and N_r  parameters are very important for the training results of PI-FL, and further analysis should be done on how to select the appropriate N_p and N_r  parameters.

**Questions:**

1.The convergence analysis about PI-FL is missing.
2. No runtime results are given in the experiments. Does PI-FL actually bring any speedup over other pFL methods? Are the actual computational and communication costs of PI-FL in an acceptable range?

---

> ### Author Rebuttal · Authors · 2023-08-08
>
> We extend our sincere appreciation to the reviewer for providing their valuable comments. We would like to address any misunderstandings and concerns regarding our work.
>
> To address the reviewer's concerns about the convergence speed, we have uploaded new evaluations (Figures 1 and 2), which demonstrate that our PI-FL algorithm achieves faster convergence and higher average personalized accuracies compared to other algorithms. It is important to emphasize that PI-FL employs the same number of communication rounds per training epoch as other clustering-based pFL algorithms, such as FedSoft. This is primarily due to the efficient piggybacking of client preference requests on model updates after aggregation.
>
> Furthermore, in terms of computation costs, PI-FL vastly outperforms clustering-based pFL algorithms. The significant advantage arises because FedSoft necessitates an additional update step during training, referred to as proximal update. On the other hand, PI-FL clients only require a single-shot personalization using cluster models, which is not only efficient but also conveniently performed by each client once the training concludes. It is worth noting that this personalization step is not mandatory during training, which further adds to the efficiency of our approach. In addition, we uploaded new comparison results (Table 1) with IFCA [1] and FedEM [2] which further bolsters our claim, as the outcomes demonstrate that FedSoft performs better than IFCA and FedEM, and our PI-FL algorithm outperforms all three (IFCA, FedEM, and FedSoft) clustering pFL algorithms.
>
> While we acknowledge the prevalence of Federated Learning in textual tasks and the potential inclusion of NLP examples in our final paper, we highlight that our PI-FL algorithm is specifically designed for resource-constrained IoT and Edge devices, utilizing smaller model sizes [3,4,5,6,7,8]. For Cross-Silo Federated Learning [1], where clients have more abundant resources, BERT might be a more appropriate option due to its LLM capabilities.
>
> As for the selection of N_p and N_r, we adopted a well-established approach in [9], where exploration is given relatively larger importance in the initial rounds. To be specific, we initialize N_r as 20 percent of all clients and dynamically reduce it based on the percentage of remaining unexplored clients. This dynamic reduction ensures an optimal exploration strategy, with the minimum N_r set to 5% once all clients have been thoroughly explored. We recognize the need to present these details more explicitly in the algorithm in the final version of the paper
> In conclusion, we have uploaded additional experiments to demonstrate the robustness and efficacy of PI-FL  following the insightful comments of the reviewer. We deeply value the reviewer's feedback, and we are committed to presenting a final version of the paper that thoroughly addresses all concerns and presents our findings.
>
> We will improve the figures as per the reviewer's recommendations.
>
> [1] A. Ghosh et al. (2022). "An Efficient Framework for Clustered Federated Learning." IEEE Transactions on Information Theory, 68(12), 8076-8091. doi: 10.1109/TIT.2022.3192506.
>
> [2] Othmane MARFOQ et al. (2021). "Federated Multi-Task Learning under a Mixture of Distributions." 35th Conference on Neural Information Processing Systems (NeurIPS 2021).
>
>
> [3] Kairouz, P. et al. (2021). Advances and Open Problems in Federated Learning.
>
> [4] Prasad, V. K. et al. (2023). Federated Learning for the Internet-of-Medical-Things: A Survey. Mathematics, 11(1), 151. doi: 10.3390/math11010151.
>
> [5] Abdel-Basset, M. et al. (2022). Federated Intrusion Detection in Blockchain-Based Smart Transportation Systems. IEEE Transactions on Intelligent Transportation Systems, 23(3), 2523-2537. doi: 10.1109/TITS.2021.3119968.
>
> [6] Farris, I. et al. (2015). Federated edge-assisted mobile clouds for service provisioning in heterogeneous IoT environments. In 2015 IEEE 2nd World Forum on Internet of Things (WF-IoT) (pp. 591-596). doi: 10.1109/WF-IoT.2015.7389120.
>
> [7] Premsankar, G. et al. (2018). Edge Computing for the Internet of Things: A Case Study. IEEE Internet of Things Journal, 5(2), 1275-1284. doi: 10.1109/JIOT.2018.2805263.
>
> [8] Qin, M. et al. (2019). Power-Constrained Edge Computing With Maximum Processing Capacity for IoT Networks. IEEE Internet of Things Journal, 6(3), 4330-4343. doi: 10.1109/JIOT.2018.2875218.
>
> [9] Lai, F. et al. (2021). Oort: Efficient Federated Learning via Guided Participant Selection. In 15th USENIX Symposium on Operating Systems Design and Implementation (OSDI 21) (pp. 19-35).

---

> > ### Comment · Reviewer_mhQ3 · 2023-08-12
> >
> > Thank you for you responses. They have addressed some of my concerns about the performnace of the proposed method.

---

> > > ### Author Response · Authors · 2023-08-12
> > >
> > > We thank the reviewer for providing a response and improving our score. We have diligently taken into consideration all the questions and suggestions raised. Specifically, we have addressed the reviewer's **question 1** by presenting new evaluation results that highlight the convergence speed. Furthermore, to answer **question 2**, we have provided a comprehensive breakdown of the computation and communication costs, drawing a comparison with other algorithms. In response to **question 3**, we have elaborated on the context of the problem we are solving within the scope of this paper. Additionally, for answering **question 5**, we have provided the methodology employed for selecting N_p and N_r. As for **question 4**, we assure the reviewer that we are committed to revising the figures as per their recommendations for the final paper version.
> > >
> > > To ensure we align with the reviewer's expectations, could you kindly guide us on the specific points you believe should be further emphasized?

---

### Author Rebuttal · Authors · 2023-08-09

Additional evaluation results as per the reviewer's recommendations.

---

### Comment · Area_Chair_td4u · 2023-08-21

Dear authors,

Thank you very much for the rebuttal and discussion with some reviewers. I will incorporate all your comments during the subsequent discussion period before recommending a decision on the paper.

---

### Decision · Program_Chairs · 2023-09-21

**Decision:**

Reject

**Comment:**

This paper proposes a personalized FL method that tries to increase the global model's appeal to the clients by incentivizing clients to participate. Based on the reviews and discussion, I recommend rejection of the paper. While the paper's idea and motivation are interesting and it was appreciated by the reviewers, the theoretical analysis and convergence guarantees have limitations. Considering other fairness measures, and using more realistic experimental settings are also directions to consider. Finally, the writing and presentation could be improved.